

# NLRP3 inflammasome activation mediates sleep deprivation-induced pyroptosis in mice

Kun Fan[1,*], Jiajun Yang[2,*], Wen-Yi Gong[3], Yong-Chao Pan[2], Peibing Zheng[2] and Xiao-Fang Yue[2]

[1] Department of Anesthesiology, Shanghai Sixth People's Hospital East Affiliated to Shanghai University of Medicine & Health Sciences Shanghai Jiao Tong University Affiliated Sixth People's Hospital, Shanghai, PR China
[2] Department of Neurology, Shanghai Sixth People's Hospital East Affiliated to Shanghai University of Medicine & Health Sciences Shanghai Jiao Tong University Affiliated Sixth People's Hospital, Shanghai, PR China
[3] Department of Anesthesiology, Shanghai First People's Hospital, Baoshan Branch, Shanghai, PR China
[*] These authors contributed equally to this work.

Corresponding author
Xiao-Fang Yue, klyxf@126.com

## ABSTRACT

**Background**. Sleep deprivation (SD) has many deleterious health effects, including cognitive decline, work ability decline, inadequate alertness, etc. Neuroinflammation plays an important role in sleep deprivation. However, the underlying mechanism is still unclear.

**Methods**. In the present study, we detected the activation of microglia and apoptosis of nerve cells in sleep deprivation (SD) mice model using IHC, HE staining and TUNEL apoptosis assay. RT-PCR array data were used to detect the expression of inflammatory bodies in hippocampal CA1 region after sleep deprivation, to explore how NLRP3 inflammasome regulates neuronal apoptosis and how specific signaling pathways are involved in SD-induced activation of NLRP3/pyrosis axis.

**Results**. We found the number of microglia significantly increased in SD mice, while this effect was blocked by sleep recovery. RT-PCR array data suggested that NLRP3 inflammasome, but not other inflammasomes, was obviously increased in hippocampus CA1 region after sleep deprivation. Mechanistically, we found that NLRP3 mediated the pyroptosis of neurocyte through GSDMD-dependent way , and P38 and ERK-MAPK signaling pathway is involved in SD-induced activation of NLRP3/pyroptosis axis. All these results suggested that MAPK/NLRP3 axis mediated SD-induced pyroptosis.

**Conclusion**. NLRP3 plays an important role in SD-induced neuroinflammation. Thus, NLRP3 inflammasome is expected to be a potential therapeutic target for SD-induced neuroinflammation.

## INTRODUCTION

Sleep deprivation (SD) refers to the abnormal sleep volume and abnormal behavior during sleep, as well as the normal rhythmic alternation disorder of sleep and awakening. Severe SD can lead to impaired neurogenesis, metabolic and cardiovascular problems, decreased immune function, weakened resistance, impaired memory, and disruption of

the blood–brain barrier. At present, the treatment of sleep disorders is mainly divided into drug treatment and non-drug treatment. However, there is a lack of research on the diagnosis and treatment of sleep disorders in patients, and the underlying mechanism has not been better understood.

Interleukin-1 (IL-1) and tumor necrosis factor-a (TNF-a) have been extensively studied to be involved in regulating physiological sleep (*Hurtado-Alvarado et al., 2013*), and the brain communicates with the external circulatory system through lymphoid tissue or blood–brain barrier (*Besedovsky, Lange & Born, 2012*). When glial cells in the nervous system release proinflammatory factors, the proinflammatory factors induce suppression of splenic natural killer (NK) cell activity, thereby affecting the function of the nervous system (*Katafuchi et al., 2009*). SD has been shown to cause disorders of the immune system and activation of inflammatory responses. For example, human experiments have found that the number of white blood cells, monocytes, and neutrophils in the peripheral blood of chronic sleep deprivers is increased (*Vgontzas et al., 2004*). At the same time, the expression of sleep-related inflammatory factors such as IL-1$\beta$, IL-6 and TNF-a increased to different degrees after SD (*Chennaoui et al., 2014*; *Irwin, Carrillo & Olmstead, 2010*). Recently, there is increasing evidence that SD causes neuroinflammation. However, the mechanism of SD-induced neuroinflammation has not been elucidated.

As a member of the Nod-like receptors (NLRs) family of inflammasomes, nucleotide-binding domain and leucine-rich repeat protein-3 (NLRP3) inflammasome is one of the most widely studied inflammasome receptor molecules, which consists of nucleotide-binding oligomerization domain (NACHT), apoptosis-speck-like protein (ASC) and procaspase-1 protein. Upon activation, NLRP3 recruits the adaptor molecule ASC, which in turn recruits the cysteine protease caspase-1. Caspase-1 activation subsequently induces the processing of pro-IL-1$\beta$ and pro-IL-18 into their mature secreted forms (*Zhou et al., 2018*). Studies have shown that NLRP3 inflammasome-mediated neuroinflammatory response is closely related to the occurrence and development of neurological diseases (*Shichita, Ito & Yoshimura, 2014*). NLRP3 inflammasome can act as a tissue damage receptor, which senses bioenergy deficiency, acidosis and oxidative stress. It regulates the maturation and secretion of IL-1beta and IL-18 by activating Procaspase-1 to cleave into p20 and p10 subunits, and induces cell death. This type of programmed death triggered by inflammasomes is also called pyroptosis (*Evavold & Kagan, 2019*). A characteristic sign of pyroptosis is membrane rupture, with the release of pro-inflammatory mediators such as IL-1beta and IL-18 outside the cell. In addition, studies have shown that once the NLRP3 inflammasome is activated, the expression level of GSDMD is increased, and the cleavage of GSDMD-NT is partially dependent on the activation of caspase-1 and the Asp280 amino acid site in its structural domain, and GSDMD-NT can bind to cell membrane phospholipids through its Glu15 and Leu156 amino acid sites, then destroy cell membrane and induce pyroptosis (*Rathkey et al., 2017*).

In our study, we explored the effect of SD-induced histological damage and the microglia by using mice model of SD. Also, we assessed the involvement of the NLRP3/caspase 1 inflammasome pathway in SD-induced neurological injury, and further revealed the potential mechanisms underlying the NLRP3, pyroptosis and the MAPK signaling pathway

under SD. Taken together, all results demonstrate that NLRP3 inflammasome activation mediates SD-induced pyroptosis in SD mice.

## MATERIALS AND METHODS

### Animal

Adult male C57BL/6 mice (23–25 g; $n = 10$ of each group) were obtained from Shanghai Model Organisms Center, Inc. (Shanghai, China), and housed in plexiglass cage under standard environmental conditions of temperature ($25 \pm 2$ °C) and humidity ($55 \pm 2\%$ RH). The food and water were provided *ad libitum* and the related animal experiments were approved by the Animal Care Committee at the Shanghai First People's Hospital, Baoshan Branch (No. 2020-Y-11). Animal experiments were conducted during daytime light hours.

### Sleep deprivation and recovery model

SD model was performed based on our preliminary results and previous relevant reports (*Franken et al., 1991*; *Xia et al., 2017*). In brief, SD was induced for 6 h, which began at 7 a.m. and ended at 1 p.m. Animals in the sham group were kept undisturbed in a separate room with the same light/dark cycle as the SD group. The treatment with SD was continued for 1-5 weeks. At the end of the fifth week, mice in the sleep recovery group were allowed to sleep for 24 h. SD mice were anesthetized using 10% chloral hydrate and sacrificed by cervical dislocation, the hippocampus region was isolated from the whole brain kept on ice, washed with 0.1 M phosphate-buffered saline solution and stored at $-80$ °C or stored in 4% PFA at 4 °C for further experiments. All mice were maintained under specific-pathogen free conditions and fed ad libitum a standard chow. The block experiment, SD mice were treated with or without SB203580 (1 μmol/L, inhibitor of p38) and U0126 (1 μmol/L, inhibitor of Erk).

### Hematoxylin and Eosin (HE) Staining

Hippocampus CA1 region was immersed in 4% paraformaldehyde and then embedded in paraffin. The tissue was cut into 4-mm sections and HE staining was carried out as follows: The section was stained with hematoxylin for 5 min, decolorized with 75% hydrochloric acid and alcohol solution for 30 s, stained with eosin for 5 min, and finally decolorized with 90% ethanol for 35 s.

### Immunohistochemistry (IHC)

Paraffin sections were then dewaxed and hydrated and then washed with PBS at room temperature. Heat mediated antigen retrieval with Tris/EDTA buffer pH 9.0 was performed. Then, the sections were washed with PBS three times and treated with 3% $H_2O_2$-methanol for 15 min. Immunostaining was carried out through incubation with antibodies against IBA-1 (4 μg/ml, ab48004, Abcam), NLRP3 (1:500, ab214185, Abcam) and GSDMD (1:1000, ab219800, Abcam).

### Immunofluorescence

The brain cryosections were first incubated with 70% Ethanol for 5 min at room temperature, then washed with PBS and incubated for 40 s with 40 mg/ml Proteinase

K, and blocked for 1 h at room temperature in blocking buffer (0.3% Triton X-100/10% Goat serum/phosphate buffer saline), and incubated with primary antibodies overnight at 4 °C in blocking buffer. Sections were then washed three times in blocking buffer and secondary antibodies were incubated for 1 h at room temperature in the dark. Sections were washed again, incubated with DAPI for 5 min at room temperature, washed and mounted in Citifluor (Agar Scientific).

Primary antibodies used for Immunofluorescence were as follow: NeuN (1:1000, ABN78, sigma) All secondary antibodies were also purchased from Abcam.

## TdT-mediated dUTP-biotin nick end labeling (TUNEL) staining

TUNEL staining was carried out using the In Situ Cell Death Detection Kit (Sigma, St. Louis, MO, USA). The sections were fixed in 4% paraformaldehyde for 1 h and treated with proteinase K (20 µg/ ml) for 15 min at 37 °C. Peroxidase blocking with 0.3% hydrogen peroxide in methanol, permeation with 0.1% sodium citrate in 0.1% Triton X-100 for 2 min. Then, the sections were then incubated with TUNEL solution from In Situ Cell Death Detection Kit for 1 h at 37 °C. The sections were examined using a Nikon Eclipse E6000 fluorescent microscope.

## ELISA assay

Hippocampus CA1 region tissue samples were gathered for the detection of IL-1$\beta$, IL-18 and TNF-$\alpha$ concentrations by ELISA kit (Solarbio). All the tests were carried out as the manufacturer's instructions. Each sample was tested in triplicate.

## Western Blot analysis

Hippocampus CA1 region was lysed with RIPA Buffer (Solarbio, Beijing, China), and the total protein concentration was quantified with BCA protein assay kit (Pierce, Rockford, IL, USA). A total of 30 µg of total extract proteins were separated on 10% SDS-PAGE and transferred to PVDF membranes (Roche, Basel, Switzerland). The membranes were incubated with primary antibodies against NLRP3 (1:1000, ab214185, Abcam), Caspase-1 (1 µg/ml, ab138483, Abcam), GSDMD (1:1000, ab219800, Abcam), ASC (1:1000, ab167165, Abcam), p-P38 (1:1000, ab178867, Abcam), P38 (1:1000, ab170099, Abcam), p-ERK (1:1000, ab201015, Abcam) , ERK (1:1000, ab32537, Abcam), p-AKT (1:1000, ab38449, Abcam), AKT (1:500, ab8805, Abcam) and $\beta$-actin (1:5000, ab6276, Abcam) overnight at 4 °C. The membranes were incubated with HRP-conjugated secondary anti-rabbit (1:5000) for 1 h at room temperature and then visualized by ECL kit (Solarbio). The intensity of bands was determined using software ImageJ (NIH, Bethesda, MA, USA).

## Quantitative reverse transcription (qRT-PCR)

Total RNA was extracted from hippocampus CA1 region of the control group, SD group and SD recovery group by Trizol reagent (Solarbio) as the manufacturer's instructions. Reverse transcriptional PCR was carried out with the iScripe$^{TM}$ cDNA Synthesis kit (Bio-Rad, Hercules, CA, USA). qPCR reaction was performed using an ABI 7500-Fast Real-Time PCR System (Applied Biosystem, Foster City, CA, USA) using LightCycler 480 SYBR Mix (Roche). The fold changes were calculated with the $2^{-\Delta\Delta Ct}$ method and GAPDH served as a normalizing control.

## Statistical analysis

All data are expressed as mean ± SD of three independent experiments. Statistical analyses were performed with the SPSS 19.0 statistical software (IBM, Armonk, NY, USA). Data was assessed for normality using D'Agostino and Pearson omnibus normality test. Values among multiple groups were compared via one-way ANOVA, and post hoc comparisons were conducted by the Bonferroni test or by Dunnett method if the homogeneity of variance was not met. $P < 0.05$ was considered statistically significant.

## RESULTS

### Sleep deprivation induced the increase of microglia and neuronal damage

To evaluate the effect of sleep deprivation induced histological damage, we did HE, Tunel staining and IHC staining for IBA-1, the marker of microglia, on brain sections of the hippocampus CA1 region. As shown in Figs. 1A and 1B, the average number of microglia was obviously increased in SD group compared with control group, while the average number of microglia decreased after 24 h sleep recovery. H&E staining data showed that the nuclear pyknosis was significantly increased in SD group compared with control group, while the effect was blocked by sleep recovery (Figs. 1C and 1D). Consistently, the Tunel staining data showed that the number of apoptosis cells in SD group is significantly higher than that in control group, and restored by sleep recovery (Figs. 1E and 1F). These data suggest that SD may mediate neuronal cell death through microglia activation.

### Sleep deprivation promoted the activation of NLRP3/caspase 1 pathway

The microglia play an important role in neuroinflammation, and the inflammasome, which leads to caspase-1 activation, is implicated in neuroinflammation. To determine which inflammasome pathway is involved in sleep deprivation induced neuronal damage, inflammasome RT-PCR array was performed. As shown in Fig. 2A, only NLRP3 mRNA levels were significantly increased in SD group, while NLRP1, NLRP2 and AIM2 were not significantly increased in SD group. To further confirmed the data, we measured the protein level of NLRP3, ASC and caspase-1, markers of the NLRP3 inflammasome pathway, in brain tissue. Our results showed that the protein levels of NLRP3, ASC and activated caspase-1 were higher at SD group compared to control group, and decreased by sleep recovery (Figs. 2B and 2C). Consistently, the IHC staining for NLRP3 also showed the NLRP3 obviously increased in SD group compared with control group, and restored by sleep recovery (Fig. 2D).

### Sleep deprivation induced the pyroptosis of neuronal cell

Activated caspase-1 indicates the presence of pyroptosis. Caspase-1 regulates the cleavage and maturation of the downstream inflammatory cytokines IL-1$\beta$ and IL-18. To evaluate the effect of Sleep deprivation on hippocampal inflammation response, The mRNA and protein level of IL-1$\beta$, IL-18, TNF-$\alpha$ was further confirmed in hippocampus CA1 region by RT-PCR and Elisa. As shown in Figs. 3A and 3B, the level of IL-1$\beta$, IL-18, TNF-$\alpha$ was

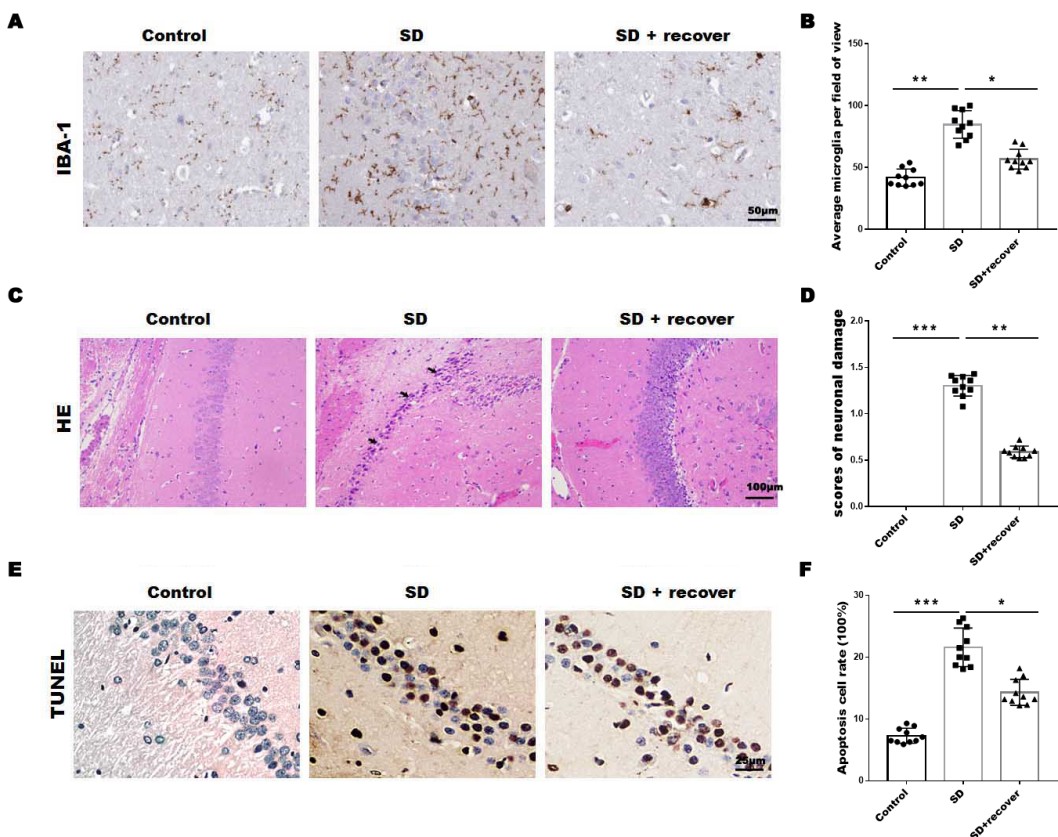

**Figure 1** **Sleep deprivation induced the increase of microglia and neuronal damage.** (A and B) IHC for IBA-1 level in hippocampus CA1 region of control group ($n = 10$), SD group ($n = 10$) and SD+ recover group ($n = 10$). Scale bars =50 µm for IHC. (C and D) H&E staining for hippocampus CA1 region in control group ($n = 10$), SD group ($n = 10$) and SD+ recover group ($n = 10$). Scale bars =100 µm for H&E. (E and F) Cell apoptosis were detected in hippocampus CA1 region in control group ($n = 10$), SD group ($n = 10$) and SD+ recover group ($n = 10$) by Tunel staining. Scale bars =25 µm for Tunel staining. Results represent the mean ofthreeindependentexperiments * $p < 0.05$. IHC, immunohistochemical; IBA-1, Ionized calcium binding adaptor molecule-1; SD, sleep deprivation; H&E staining, hematoxylin-eosin staining.

markedly increased in SD group, while the effect was blocked by sleep recovery. Besides, the protein level of pyroptosis marker GSDMD and apoptosis marker bax were analyzed by Western-blotting, the data revealed a significant increase in the expression of GSDMD and bax in the mouse brain of hippocampus CA1 region, and the expression of GSDMD, but not the expression of bax, was reversed by 24 h sleep recovery (Figs. 3C–3D). We further detected the GSDMD expression by Immunohistochemistry. As shown in Fig. 3E, the expression of GSDMD was markedly increased in SD group, while the effect was blocked by sleep recovery. To further confirmed the pyroptosis cell type, the mouse brain of hippocampus CA1 region was stained with the neuronal marker NeuN and GSDMD. As shown in the Fig. 3F, brains showed colocalization of the neuronal marker NeuN with GSDMD. These data suggested that SD-induced the pyroptosis of neuronal cell.

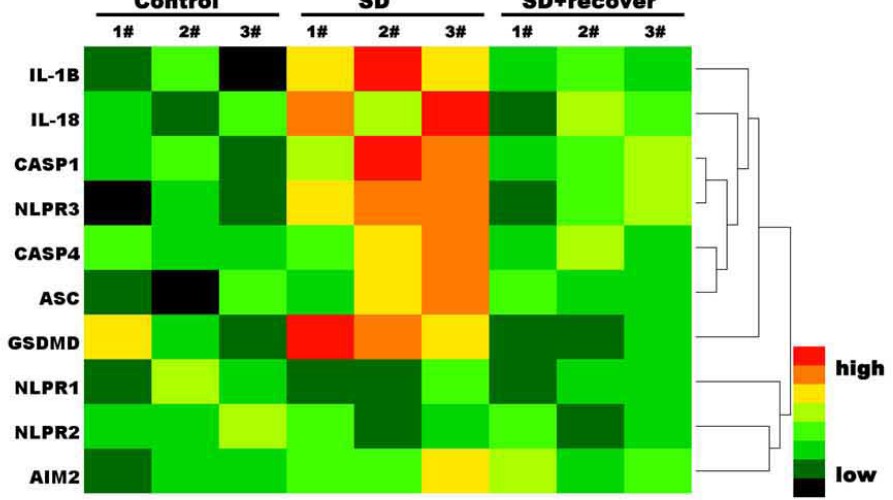

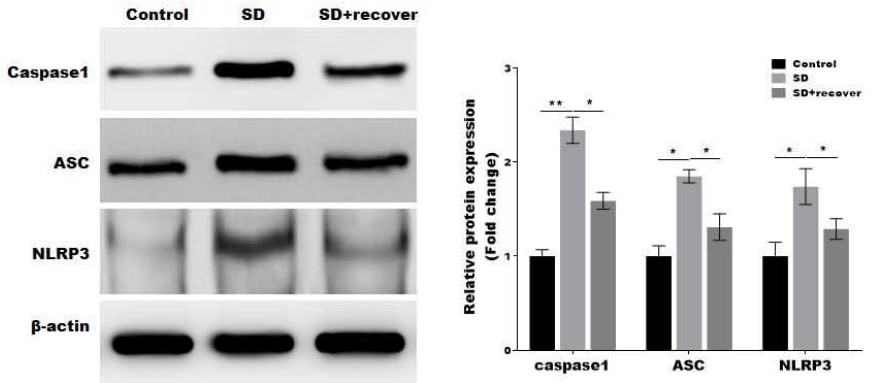

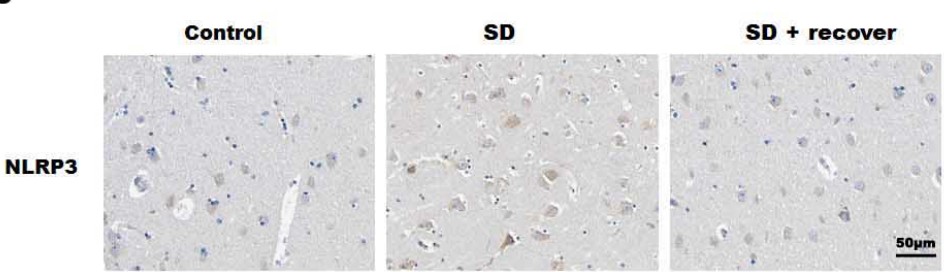

**Figure 2** **Sleep deprivation promoted the activation of NLRP3/caspase 1 pathway.** (A) RT-PCR array for the expression levels of genes (IL-1 $\beta$, IL-18, Caspase-1, NLRP3, Caspase-4, ASC, GSDMD, NLRP1, NLRP2 and AIM2) in control group, SD group and SD+ recover group ($n = 5$). (B and C) Western blot analysis for Caspase-1, ASC and NLRP3 protein level in hippocampus CA1 region of control group, SD group and SD+ recover group ($n = 5$). (D) IHC for NLRP3 level in hippocampus CA1 region of control group, SD group and SD+ recover group. Scale bars = 50 μm for IHC ($n = 5$). Results represent the mean of three independent experiments $^*p < 0.05$, $^{**}p < 0.01$. RT-PCR, Quantitative real-time PCR; IHC, immunohistochemical; SD, sleep deprivation; ASC, -associated speck-like protein containing a CARD.

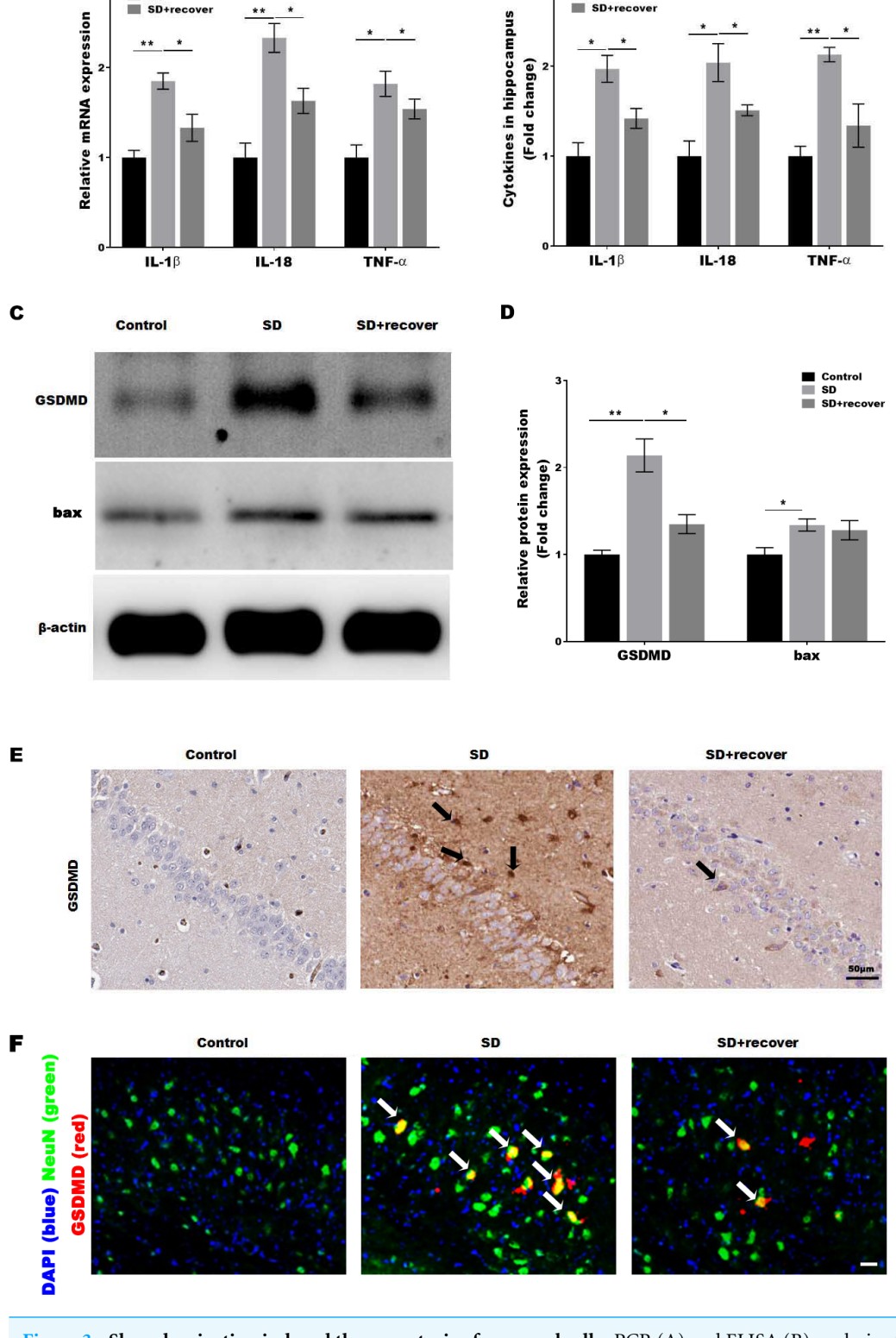

**Figure 3  Sleep deprivation induced the pyroptosis of neuronal cell.** qPCR (A) and ELISA (B) analysis of IL-1β, IL-18 and TNF-α expression in hippocampus (continued on next page...)

**Figure 3 (…continued)**
CA1 region of control group, SD group and SD+ recover group and qPCR analysis of IL-1 $\beta$, IL-18 and TNF-$\alpha$ expression in hippocampus of control group, SD group and SD+ recover group ($n = 5$). (C and D) Western blot analysis for GSDMD and bax protein level in mouse hippocampus CA1 region of control group, SD group and SD+ recover group ($n = 5$). (E) IHC for GSDMD level in mouse hippocampus CA1 region of control group, SD group and SD+ recover group ($n = 5$). Scale bars = 50 $\mu$m for IHC. (F) Fluorescence co-localization for GSDMD and NeuN in mouse hippocampus CA1 region of control group, SD group and SD+ recover group ($n = 5$). Scale bars = 20 $\mu$m. Results represent the mean of three independent experiments *$p < 0.05$, **$p < 0.01$. qPCR, Quantitative real-time PCR; IHC, immunohistochemical; SD, sleep deprivation.

## Sleep deprivation induces the activation of P38 and ERK MAPKs pathway

It is well-known that the MAPKs and AKT activation is related to the of NLRP3 and cell pyroptosis (*Zhang et al., 2020*; *Zhou et al., 2019*). Thus, we detected the activation of MAPKs pathways in control and SD model with or without 24 h sleep recovery brain tissue of hippocampus CA1 region. Intriguingly, we found that phosphorylation of p38 and ERK1/2 was increased after sleep deprivation but not AKT phosphorylation compared with the control group, while sleep recovery significantly inhibited p38 and ERK1/2 phosphorylation (Figs. 4A–4F). To further confirmed the p38 and ERK1/2 pathway in the SD-induced NLRP3 and pyroptosis activation, the SD mice were treated with or without the p38 and ERK pathway inhibitor SB203580 and U0126, and the NLRP3 and pyroptosis activation was analyzed by Western blot. As shown in Figs. 4G and 4H, the p38 inhibitor SB203580 and the ERK inhibitor U0126 could significantly block the SD-induced activation of NLRP3 and neuronal pyroptosis.

## DISCUSSION

Sleep deprivation (SD) is very common, with 20% of adults reported to be sleep deprived (*Bandyopadhyay & Sigua, 2019*). SD has many harmful effects, including increased risk of stroke, adiposity, glycuresis, tumor, permanent cognitive impairment, osteoporosis, cardiovascular disease and mortality (*Tobaldini et al., 2017*). Increasing evidence suggest that SD causes neuroinflammation in the brain (*Xue et al., 2019*). However, the specific mechanism by which SD triggers neuroinflammation in the brain is unclear. In our study, we found that (1) SD-induced the increase of microglia and neuronal damage; (2) SD promoted the activation of NLRP3/caspase 1 pathway; (3) SD-induced the pyroptosis of neuronal cell; (4) SD induces the activation of P38 and ERK MAPKs pathway. The current data verify the NLRP3 inflammasome activation mediates SD-induced pyroptosis in mice.

Increasing evidence confirms that SD can trigger neuroinflammation (*Wadhwa et al., 2019*). Microglia, as a kind of innate immune cells in the central nervous system, are the key links in the process of inflammation (*Su et al., 2016*). Microglia can be activated either directly by some toxic substances or endogenous proteins or indirectly by dying neurons. Long-term activation of microglia can lead to neuronal damage (*Fakhoury, 2015*; *Suzumura, 2014*). Activated microglia, on the one hand, exert the role of phagocytes in the brain through direct contact with neurons; on the other hand, secrete some inflammatory mediators such as NO, TNF-a, IL-1 and chemokines. Overactivation of microglia enlarges

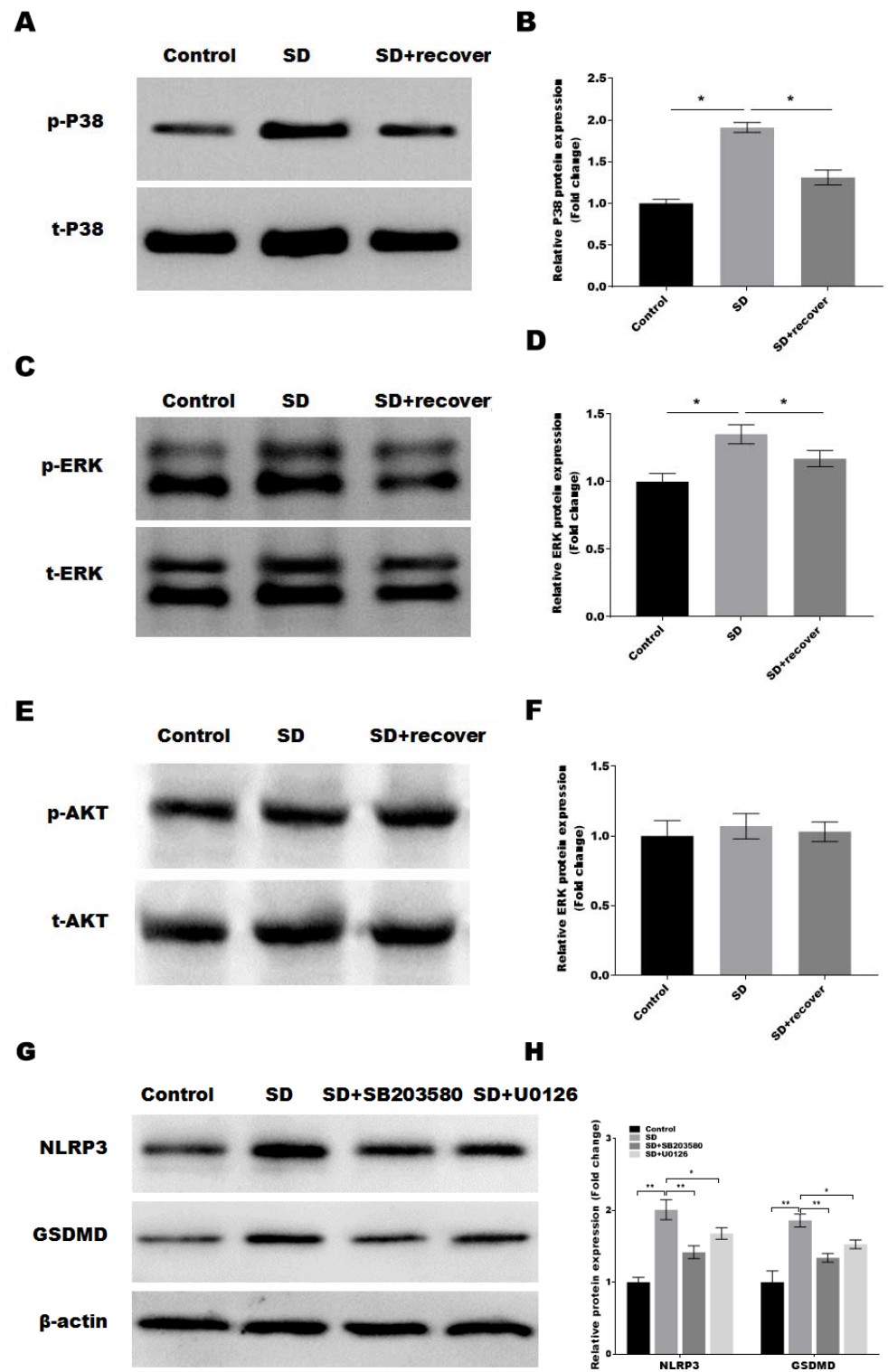

**Figure 4** Sleep deprivation induces the activation of P38 and ERK MAPKs pathway. 

**Figure 4 (...continued)**
(A and B) Western blot analysis for p-P38 and P38 protein level in mouse hippocampus CA1 region of control group, SD group and SD+ recover group ($n = 5$). (C and D) Western blot analysis for p-ERK and ERK protein level in mouse hippocampus CA1 region of control group, SD group and SD+ recover group ($n = 5$). (E and F) Western blot analysis for p-AKT and AKT protein level in mouse hippocampus CA1 region of control group, SD group and SD+ recover group ($n = 5$). (G and H) Western blot analysis for NLRP3 and GSDMD protein level in mouse hippocampus CA1 region of control group, SD group, SD+ SB203580, and SD+ U0126 group ($n = 5$) Results represent the mean of three independent experiments * $p < 0.05$. SD, sleep deprivation.

the inflammatory response continuously and produces a large number of cytokines and reactive oxygen species products, which further aggravates neuronal injury. In our study, we constructed SD mice model to detect the expression of IBA-1 (the marker of microglia) in the brain part of hippocampal CA1 region of SD mice by IHC. The results showed that the expression level of IBA-1 was significantly up-regulated in the hippocampal CA1 region after SD and deceased after 24 h sleep recovery, suggesting that SD promoted the activation of microglia. Then, the level of IL-1 $\beta$, IL-18, TNF-$\alpha$ (inflammatory factor) was further detected in hippocampus CA1 region of SD mice. The results showed that the expression of IL-1$\beta$, IL-18, TNF-$\alpha$ was significantly increased after SD and deceased after 24 h sleep recovery, suggesting that SD triggers inflammation by activating microglia, leading to neuronal damage.

Pyroptosis is a recently defined form of cell death that differs from other forms of cell death and is characterized by Caspase-1 activation, cell membrane pore formation, and release of cellular contents (*Fang et al., 2020*). Although a certain degree of pyroptosis can endow the host with a mechanism of defense against infectious diseases, excessive pyroptosis is harmful (*Vande Walle & Lamkanfi, 2016*)). NLRs act as an intracellular pattern receptor that can activate Caspase-1 and NF-$\kappa$B, MAPK signaling pathways, promote the production of proinflammatory factors, and thus initiate innate and acquired immunity after recognizing the corresponding ligands (*Liu et al., 2019*). NLRP3 is an important member of the NLR family, activated caspase-1 leads to differences in the formation of plasma membrane pores within and outside cells (*Fink & Cookson, 2006*). High intracellular osmotic pressure leads to tissue fluid entering cells, resulting in swelling and dissolution of cells and release of inflammatory cytokines (IL-1$\beta$, IL-18, and TNF-$\alpha$) (*Miggin et al., 2007*). *Zielinski et al., (2017)* found that the activation of the NLRP3 inflammasome can modulate sleep induced by both increased wakefulness and a bacterial component in the brain. Here, we examined several inflammasome-related genes by RT-PCR array to determine which inflammasome pathways are involved in neuronal damage induced by SD, and the results showed that only NLRP3 mRNA levels were significantly increased in SD group and the protein levels of the markers of the NLRP3 inflammasome pathway (NLRP3, ASC and caspase-1) were increased, while decreased by sleep recovery. Additionally, the level of IL-1$\beta$, IL-18, TNF-$\alpha$ was also significantly increased in SD group. This is consistent with the findings of *Xia et al. (2017)*, who found that in SD group, SD stimulated formation of NLRP3 inflammasomes, with subsequent induction of the maturation of IL-1$\beta$ and IL-18.

GSDMD is a member of the gasdermins (GSDMs) family, a common substrate of caspase-1 and caspase-4/5/11, and an executor of pyroptosis. GSDMD consists of two conserved domains: The C-terminal inhibitory domain (21ku) and the N-terminal effector domain (32ku), where the N-terminal is cytotoxic and can bind to lipid components to form holes in the cell membrane. GSDMD can be cleaved into C-terminal and N-terminal by caspase-1/4/5/11, and its N-terminal oligomerizes on the cell membrane to form non-selective pore channels, releasing mature IL-18 and IL-1 $\beta$, and inducing the occurrence of pyroptosis (*He et al., 2015*; *Zhaolin et al., 2019*). In our study, the expression of GSDMD was detected in hippocampus CA1 region of SD group and 24 h sleep recovery group by Western Blot and IHC assay, the data revealed a significant increase in the expression of GSDMD in the mouse hippocampus CA1 region of SD group, and this effect was reversed by 24 h sleep recovery. In addition, the proptosis marker GSDMD colocalization with the neuronal marker NeuN, suggesting that NLRP3 mediates the pyroptosis of neurocyte through GSDMD-dependent way. To further investigate the mechanism of pyroptosis, we attempted to identify the signaling pathways that regulate pyroptosis. LI et al. confirmed that blockage of p38 MAPK signaling pathway with SB203580 suppressed macrophage pyroptosis and LPS-induced acute lung injury through negative regulation of NLRP3 inflammasome activation. In addition, DHA ameliorated I/R-induced injury by inhibiting pyroptosis of hepatocytes induced in liver I/R injury in vivo and in vitro through the PI3K/Akt pathway. In our study, we detected the activation of MAPKs pathways in control and SD model with or without 24 h sleep recovery brain tissue of hippocampus CA1 region, phosphorylation of p38 and ERK1/2 was increased after sleep deprivation but not AKT phosphorylation compared with control group, while sleep recovery significantly inhibited p38 and ERK1/2 phosphorylation. All these results suggest that P38 and ERK-MAPK signaling pathway is involved in SD-induced activation of NLRP3/pyroptosis axis.

At present, our research still has some limitations. The specific mediating mechanism of P38 and ERK-MAPK signaling pathway is involved in SD-induced activation of NLRP3/pyrosis axis remains unclear and needs further exploration. In addition, we found that most of the pyroptosis occurred in the hippocampus, and next we are going to explore the effects of SD on cognitive impairment in animals.

## CONCLUSIONS

The results of our study demonstrate that MAPK/NLRP3 axis may play a vital role in neuronal pyroptosis in the development of SD. These results provide a new direction for the clinical treatment of SD.

### Funding

This work was supported by the National Natural Science Foundation of China (grant number 31400953). The funders had no role in study design, data collection and analysis, decision to publish, or preparation of the manuscript.

## Grant Disclosures

The following grant information was disclosed by the authors:
National Natural Science Foundation of China: 31400953.

## Competing Interests

The authors declare there are no competing interests.

## Author Contributions

- Kun Fan, Jiajun Yang, Wen-Yi Gong, Yong-Chao Pan, Peibing Zheng and Xiao-Fang Yue conceived and designed the experiments, performed the experiments, analyzed the data, prepared figures and/or tables, authored or reviewed drafts of the paper, and approved the final draft.

## Animal Ethics

The following information was supplied relating to ethical approvals (i.e., approving body and any reference numbers):

The Animal Care Committee at the Shanghai First People's Hospital, Baoshan Branch approved this research (No. 2020-Y-11).

## Data Availability

The raw data for Western Blot, qPCR and ELISA are available in the Supplementary File.

## Supplemental Information

Supplemental information for this article can be found online at http://dx.doi.org/10.7717/peerj.11609#supplemental-information.

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
