# Peer review of "NLRP3 inflammasome activation mediates sleep deprivation-induced pyroptosis in mice"

_PeerJ, doi:10.7717/peerj.11609_

## Round 0.1 · original submission · Major Revisions

Please pay particular attention to the reviewers' comments regarding the experimental design (in particular the statistical analysis of results and methodological details) and the issues raised in relation to the recovery of neurons after a 24h period (this is a critical aspect that must be clarified by the authors).

Reviewer 1 ·

Basic reporting

The authors investigated the mechanisms underlying sleep deprivation-(SD) induced neuroinflammation which have not been completely elucidated yet. For that purpose, microglia activation was investigated in a mouse model of SD and the stimulation of the NLRP3/caspase 1 inflammasome pathway was determined. Noteworthy, the authors show that SD-induced the NLRP3 inflammasome components and the release of cytokines typical of pyroptosis. Also, SD triggered an overexpression of GSDMD, a pore-forming protein associated to pyroptosis.
It would be interesting to determine whether pyroptosis occurs in microglia and whether different cell death programs induced by SD could occur in distinct cell types.

1. The work will benefit if the authors describe in more detail the structure of the NLRP3 inflammasome. In the introduction p. 5 lines 66-68 there is a mention to the NACHT domain of the platform protein NLRP3 but other protein domains in both NLRP3 and ASC and procaspase-1 contribute to the assembly of the NLRP3 inflammasome.

2. The results support that SD may induce NLRP3 inflammasome activation. Within this context I suggest the inclusion of the article “The NLRP3 inflammasome modulates sleep and NREM sleep delta power induced by spontaneous wakefulness, sleep deprivation and lipopolysaccharide” doi:10.1016/j.bbi.2017.01.012 in the references of the manuscript.

3. The number of animals used in each study should be stated in the legend of figures 2-4 as it is in figure 1.

4. In the Introduction, p. 5 line 73 please substitute P20 and P10 subunits by p20 and p10 subunits.

5. I suggest the Methods subsection of Hematoxylin and Eosin (HE) Staining should appear before the Immunohistochemistry (IHC) subsection since the description that the hippocampus CA1 region was immersed in 4% paraformaldehyde and then embedded in paraffin is indicated in the subsection of Hematoxylin and Eosin (HE) Staining.

6. The bar graphics regarding in vivo studies would be more informative if substituted by a type of graph were we could see the distribution of the values for each animal, such as an aligned dot plot.

7. The manuscript needs a thorough English Language review.

Experimental design

1. The post-hoc test associated to the one-way analysis of variance (ANOVA) that was used in the statistical analysis of the results should be indicated in the methods section. Moreover, since Both ANOVA and Student’s t test assume data normality the authors should indicate whether the results pass a normality test (D’Agostino & Pearson omnibus normality test or the Shapiro-Wilk normality test). Otherwise the statistical analyses should be performed using suitable non-parametric tests like the Kruskall-Wallis test or the Mann-Whitney test.

2. It should be clarified in the manuscript whether the ELISA kits used for the detection of cytokines (figure 3), namely IL-1β and IL-18 detect the mature form of the cytokine or its pro-form. This is relevant regarding the conclusion that SD-induced NLRP3/caspase-1 pathway activation.

Validity of the findings

1. Results section, page 9: The authors evaluate the effect of SD in neurodegeneration through the analysis of nuclear pyknosis and the TUNEL assay (Figure 1). It was observed that SD-induced nuclear pyknosis and apoptosis of neuronal cells. The SD protocol comprises a period of 6 hours sleep deprivation sustained for the 1-5 weeks.
1. 1. Which was the basis to select a period of 24 h as a recovery interval at the end of the 5th week of continued SD? Please indicate the rational in the manuscript.
1.2 A 24 h recovery period at the end of the SD protocol may inhibit SD-induced apoptosis in neuronal post-mitotic cells as concluded by the authors. Please give an explanation for this observation in the manuscript as well as for the decrease in the number of microglia.
1.3 The authors indicate that SD induces pyroptosis which is a cell death program distinct from apoptosis. Therefore the authors should better clarify with adequate markers whether SD-induced cell death regards apoptosis, pyroptosis or both mechanisms. Also, it would be interesting to understand whether SD-induced cell death in different cell types could be mediated by different mechanisms.

2. The authors investigated whether SD promoted the activation of NLRP3/caspase 1 pathway. As shown in Figure 2A, the NLRP3, Caspase-1, ASC as well as caspase-4 mRNA levels were significantly increased in SD group. Caspase-4 mediates non-canonical activation of the NLRP3 inflammasome (as shown in the article “Caspase-4 mediates non-canonical activation of the NLRP3 inflammasome in human myeloid cells” doi: 10.1002/eji.201545523) and GSDMD is a substrate of caspase-4. It would be interesting to discuss the possible contribution of caspase-4 for SD-induced pyroptosis.

3. In Figure 3 E regarding IHC for GSDMD level in mouse hippocampus CA1 region it would be helpful to indicate with an arrow the immunostaining for GSDMD. Indeed I am not sure whether GSDMD staining corresponds to the purple or the strong brown staining observed in the mice submitted to SD. Also, the color on the representative control image is very different even from the experimental condition SD+recovery, which according to the results is expected to be similar to control.

4. In the discussion, page 11 line 216, it is stated that SD induced the pyroptosis of neuronal cell. However the markers of pyroptosis were analysed in tissue homogenates which does not allow for the conclusion that pyroptosis occurred in neuronal cells. The only pyroptosis marker analysed by IHC was GSDMD (figure 3). The study of co-localization with a neuronal and/or microglia marker would be important to clarify this issue.

Reviewer 2 ·

Basic reporting

1. Using a mouse model of sleep deprivation (SD) the authors showed SD-induced NLRP3 activation and pyroptosis. Although an interesting research topic, this is not novel. Recent published studies already demonstrated that SD activates NLRP3 inflammasome activity in mouse neurons and astrocytes (Niznikiewicz et al., 2017; Smith et al., 2019; Zielinski et al., 2020) highlighting the potential of NLRP3 as a therapeutic target (Smith et al., 2019). These research works should be discussed. In this work, the authors suggest p38 MAPK signaling pathway mediates SD-induced activation of NLRP3 and pyroptosis. The authors should give more relevance to this result and better explore this mechanism in order to increase the novelty and the relevance of their work.

2. The English language should be revised and improved throughout the article. There are some incomplete sentences.

Experimental design

3. How the brains were fixed? Post-fixation or brain perfusion? This is not clearly mentioned in the methods section.

4. To perform the statistical analysis the authors used one-way ANOVA and Student’s t-test. Do the results follow the normality?

Validity of the findings

5. I would suggest blocking the p38 MAPK signaling pathway with SB203580 in order to confirm the involvement of this pathway in SD-induced pyroptosis and NLRP activation and increase the validity of their findings.

Annotated reviews are not available for download in order to protect the identity of reviewers who chose to remain anonymous.

---

## Round 0.2 · Minor Revisions

Your paper has been improved.
However, I still have doubts about the recovery of neuronal cells death. It is shown that sleep deprivation increases cell death and that 24hr of sleep decreases cell death. I request the authors to clearly explain the decrease in cell death after 24h recovery; this period of time is short for neurogenesis.

Reviewer 2 ·

Basic reporting

The authors clearly improved the manuscript, but I would suggest to carefully revise the English writting. There are still meaningless phrases.
For exemple: To further confirmed the p38 and ERK1/2 pathway in the SD induced NLRP3 and pyroptosis activation, the SD mice treated with or without the p38 and ERK pathway inhibitors, SB203580 and U0126, and the NLRP3 and pyroptosis activation was analyzed by Western blot.Should it be "the SD mice were treated with or without the p38 ..."?

Experimental design

The authors improved the manuscript.

Validity of the findings

The authors improved the manuscript.

---

## Round 0.3 · accepted · Accept

I would like to thank the authors for their efforts in improving the paper.